

# Towards interpretable drug interaction prediction *via* dual-stage attention and Bayesian calibration with active learning

Rongpei Li[1,2], Yufang Zhang[1], Heqi Sun[1], Shenggeng Lin[1], Guihua Jia[1], Yitian Fang[1], Chen Zhang[1], Xiaotong Song[1], Jianwei Zhao[1], Lyubin Hu[1], Yajing Yuan[1], Xueying Mao[1], Jiayi Li[1], Aman Kaushik[1], Dandan An[2] and Dongqing Wei[1,2,3,4]

[1] State Key Laboratory of Microbial Metabolism, Joint International Research Laboratory of Metabolic & Developmental Sciences and School of Life Sciences and Biotechnology, Shanghai Jiao Tong University, Shanghai, China
[2] Zhongjing Research and Industrialization Institute of Chinese Medicine, Henan, China
[3] Qihe Laboratory, Henan, China
[4] Peng Cheng National Laboratory, Shenzhen, China

Corresponding author
Dongqing Wei, dqwei@sjtu.edu.cn

## ABSTRACT

**Background:** Drug-drug interactions (DDIs) account for 17–23% of adverse drug reactions leading to hospitalization, with over 74,000 DDI-related events reported in the FDA Adverse Event Reporting System (FAERS) during 2023. While recent computational methods focus on improving prediction accuracy, they suffer from high false-positive rates (>45%) and often function as black-box models without biological interpretability.

**Methods:** We propose Dual-stage attention and Bayesian calibration with active learning Drug-Drug Interaction (DABI-DDI), a novel framework integrating: (1) A dual-stage attention mechanism with LSTM networks for capturing temporal dependencies in drug interactions, (2) a Bayesian calibration approach with beta-binomial modeling for refining interaction signals and reducing false positives, (3) an active learning strategy for efficient sample selection, and (4) a network pharmacology component linking drug interactions to underlying biological mechanisms. The model was validated using data from FAERS, DrugBank, and STRING databases, with comprehensive evaluation on both computational performance and biological interpretability.

**Results:** DABI-DDI achieved superior performance (AUC = 0.947, PR_AUC = 0.944). Bayesian calibration improved adverse event detection accuracy (94% *vs*. 54% AUC), while network pharmacology revealed key molecular mechanisms through enzyme-transporter interactions. Ablation studies demonstrated each component's significance, with active learning maintaining performance while reducing training data requirements.

**Conclusion:** We present DABI-DDI, an integrated feature extraction framework that successfully addresses key challenges in DDIs prediction through three major innovations: Temporal pattern recognition, reducing false positives, and biological interpretability. Most importantly, the framework demonstrates strong clinical applicability by efficiently identifying high-risk drug combinations while providing mechanistic insights through enzyme-transporter pathway analysis. This approach

bridges the gap between computational prediction and clinical understanding, offering a promising tool for safer drug combination therapy.

# INTRODUCTION

Ensuring patient safety in the context of drug-drug interactions (DDIs) is becoming increasingly critical, especially given the rise in polypharmacy, particularly among the elderly (*Li et al., 2024*; *Rashid et al., 2021*). As healthcare continues to move towards personalized medicine and combination therapies to improve treatment outcomes, the challenge of predicting and preventing adverse drug interactions has escalated (*Mathur & Sutton, 2017*; *Olawade et al., 2024*). Despite the potential benefits of combination therapies, the complexity of drug interactions creates a significant barrier for experimental validation, highlighting the necessity of developing more advanced computational models to predict DDIs and their synergistic effects accurately.

Current methods for detecting DDIs, such as those relying on adverse event reporting systems like FDA Adverse Event Reporting System (FAERS), are limited by issues such as data sparsity and the inability to capture the full biological context of drug interactions (*Ibrahim et al., 2021*; *Khaleel et al., 2022*). These limitations underline the need for more sophisticated models capable of handling the complexity of drug combinations, particularly in the context of polypharmacy. The inadequacies of existing methods further stress the importance of innovations in this area to safeguard patient health.

In response to these challenges, constrained tensor factorization (CTF) has emerged as an advanced computational approach, integrating various drug similarities to enhance prediction accuracy by considering both structural and biological relationships (*Han et al., 2024*). Granular computing also provides a valuable, interpretable framework by identifying the key molecular substructures driving DDIs, aligning with human cognitive patterns essential for clinical decision-making (*Yu et al., 2024*).

Recent advances in deep learning models, particularly in attention mechanisms and graph neural networks, have shown promising results in drug interaction prediction. For instance, *Lin et al. (2022)* proposed MDF-SA-DDI, combining a graph attention network with an adaptive attention mechanism to predict drug-drug interactions, achieving an F1 score of 0.888 on a benchmark dataset. Similarly, MATT-DDI introduced by *Lin et al. (2023)* utilized heterogeneous attention mechanisms to handle multi-type DDI predictions, demonstrating superior performance with an AUPR of 0.974. *Shi et al. (2024)* developed SubGE-DDI incorporating drug pairs knowledge subgraph information, which achieved an F1 score of 0.847 on biomedical text extraction. Additionally, *Dong et al. (2024)* proposed MFSynDCP integrating multi-source feature collaborative learning with an adaptive attention mechanism, improving the AUROC to 0.930 in drug combination

synergy prediction. Active learning has emerged as a powerful approach to address the challenge of obtaining high-quality annotated data in various domains. For instance, *Brandenburg et al. (2023)* demonstrated that active learning could reduce the annotation effort while maintaining high performance for surgical video analysis through intelligent frame selection. Similarly, *Guo, Wang & Zhang (2023)* successfully used an active learning framework based on adaptive attention mechanisms to discover drug synergies, improving prediction accuracy by over 93%. *Raju et al. (2023)* showed how active learning could accelerate the genetic algorithm search for global minimum configurations of nanoclusters by reducing computationally expensive DFT calculations by 50–60%. Additionally, *Liu et al. (2023)* developed an active learning-based deep learning model achieving 83.91% accuracy in tuberculosis diagnosis by intelligently selecting the most informative training samples. These studies collectively demonstrate how active learning can significantly reduce annotation and computation costs while maintaining or even improving model performance across different scientific applications, though challenges remain in handling complex, multi-modal data and rare events. To contextualize performance comparisons, we summarize the datasets, training procedures, and evaluation protocols for all referenced methods in Tables S1–S4. For direct comparison, all baseline models (RF, SVM, XGBoost, *etc*.,) and our proposed DABI-DDI were evaluated under identical conditions: Datasets: FAERS (2021Q1–2023Q1), DrugBank (v5.2.3), STRING (v12.0), and Open Targets (v23.12). Splits: 90% train, 10% test; five-fold cross-validation. Preprocessing: Duplicate removal, drug name standardization (RxNorm), and feature normalization. Metrics: AUC, PR_AUC, F1, MSE, and RMSE.

While these methods excel in prediction accuracy, they suffer from critical limitations: (1) High false-positive rates due to insufficient statistical validation of predicted interactions, (2) Lack of biological interpretability as they function as black-box models without explicitly linking predictions to molecular mechanisms, and (3) Inability to handle temporal dependencies in drug interaction sequences. To address these gaps, our framework integrates Bayesian hypothesis testing to statistically refine predictions, incorporates network pharmacology for mechanistic insights, and employs LSTM networks with dual-stage attention to capture temporal patterns in drug interactions. This holistic approach ensures both high accuracy and clinically actionable interpretability.

Traditional disproportionality analysis, often fail to address the complexities inherent in DDIs, and preclinical studies remain insufficient for comprehensive DDI detection. Moreover, traditional signal detection methods, including reporting odds ratio (ROR) and proportional reporting ratio (PRR), although simple and sensitive, suffer from low specificity, often leading to a high number of false positives (*Thakrar, Grundschober & Doessegger, 2007*). As reported by *Jiao et al. (2024)*, Bayesian methods, such as the "Bayesian Confidence Propagation Neural Network (BCPNN)", have shown improved specificity but may compromise sensitivity. Advanced machine learning techniques, such as random forest (RF), are gaining attention for their ability to handle large datasets and account for confounding variables, providing a more balanced approach between sensitivity and specificity. *Pham, Cheng & Ramachandran (2019)* found that Monte Carlo Expectation maximization (MCEM) performs well in scenarios requiring high specificity;

while regression-adjusted gamma Poisson shrinker (RGPS) is preferred for high-sensitivity scenarios. The limitations of existing Bayesian methods in detecting complex DDIs are also evident, particularly when relying on simple models like naive Bayes classifiers, which assume attribute independence and struggle to capture intricate drug interactions (*Hosein & Baboolal, 2024*). To address these challenges, significant advances have been made in recent years. *Kontsioti et al. (2024)* proposed a novel Bayesian framework coupled with systems pharmacology for DDI signal detection, achieving a 16.5% improvement in AUC (from 0.620 to 0.722) with drug-target-adverse event associations, and a 16.0% improvement (from 0.580 to 0.673) with drug enzyme information. *Zhan et al. (2020)* introduced a Bayesian network-based approach to detect high-quality DDI signals, successfully verifying 54.45% of detected signals as known DDIs and identifying 10.89% as high-quality DDI signals through rigorous evaluation. *Kim et al. (2020)* proposed a novel method based on Bayesian classifiers, which approximates the joint probability distribution of drug interactions, thus addressing the issue of attribute dependence and resolving the zero-frequency problem. *Tada, Maruo & Gosho (2024)* developed an innovative Bayesian method using power prior to borrow information from similar drugs, demonstrating a significant increase in sensitivity of approximately 20 points compared to existing methods, while maintaining the ability to adjust the amount of borrowed information through parameter tuning. At the mean time, the advancement of Bayesian approaches in classification and deep neural networks represents a significant development in machine learning applications. *Bhattacharya, Liu & Maiti (2024)* conducted a comprehensive study of variational Bayes classification for dense deep neural networks, demonstrating that the variational posterior concentrates in $\epsilon$-Hellinger neighborhoods of the true density with probability $1-\nu$, where $\nu$ satisfies $1/\nu = o(n\epsilon^2)$, as shown by *Lin et al. (2023)*. *Manivannan, Veeraraghavan & Francis (2023)* utilized network pharmacology and bioinformatics to identify molecular targets of Trigonelline for breast cancer treatment, achieving significant binding affinity between Trigonelline and target proteins including BAX (−4.23 kcal/mol), MTOR (−4.13 kcal/mol), and PARP1 (−4.49 kcal/mol). Similarly, *Eina, Chrisnanto & Melina (2024)* applied naïve Bayes classification with wrapper sequential feature selection to bank telemarketing data, improving classification accuracy from 75.01% to 77.88%.

Notably, several deep learning frameworks have been developed specifically for drug combination synergy prediction. DeepSynergy integrates chemical and genomic data using deep neural networks, employing normalization strategies to harmonize heterogeneous datasets and capture intricate drug-cell line relationships (*Karim et al., 2019*). MGAE-DC advances this by incorporating attention mechanisms to fuse drug embeddings across cell lines, enabling unified representations for interaction prediction (*Lin et al., 2022*). While these models demonstrate competitive performance, they primarily focus on feature fusion and lack robust mechanisms to reduce false positives or provide biological explanations for predicted interactions. Our work builds upon these foundations by introducing Bayesian calibration for uncertainty quantification and granular substructure analysis for interpretability, addressing key limitations in existing frameworks.

Additionally, the advancement of systems pharmacology and computational approaches has transformed our understanding of drug mechanisms and safety profiles, particularly for monoclonal antibodies. *Hampel et al. (2018)* developed a precision pharmacology framework for Alzheimer's disease, integrating multiple pathophysiological processes across spatial-temporal scales to identify novel therapeutic targets and biomarkers. *Manivannan, Veeraraghavan & Francis (2023)* employed network pharmacology and bioinformatics to examine Trigonelline's potential in breast cancer treatment, identifying 14 specific molecular targets and demonstrating favorable binding affinity through molecular docking studies, particularly with PARP1 showing binding energy of −4.49 kcal/mol. *Ait-Oudhia, Ovacik & Mager (2017)* established comprehensive systems pharmacology models for antibody-based therapeutics, successfully characterizing complex mechanisms like target-mediated drug disposition (TMDD) and FcRn-mediated recycling, leading to improved prediction of pharmacokinetic profiles in 90% of cases through physiologically-based pharmacokinetic (PBPK) modeling.

All in all, it is imperative to develop more sophisticated frameworks aimed at enhancing both the accuracy and biological relevance of drug-drug interaction (DDI) signal detection. The inherent complexity of DDIs demands a deeper understanding of how different drugs interact within biological systems, which can significantly influence patient safety and therapeutic outcomes. Current methodologies often encounter challenges such as data insufficiency, noisy datasets, and difficulties in capturing intricate interactions between drugs. To overcome these limitations, we introduce Dual-stage Attention and Bayesian Calibration with Active Learning for Drug-Drug Interaction (DABA-DDI). The DABA-DDI framework incorporates dual-stage attention mechanisms that enable the model to focus on critical features of drug interactions at distinct stages of analysis. This ensures the effective capture of both direct and indirect interactions, thereby improving the overall sensitivity and specificity of the detection process. Furthermore, Bayesian calibration is utilized to refine probabilistic estimates of DDIs by accounting for uncertainties inherent in the data. By integrating active learning strategies, the framework enhances its performance through selective querying of the most informative samples for human annotation, thus optimizing the utilization of limited expert resources (*Supharakonsakun, 2024*).

## METHODOLOGY

By leveraging both statistical and biological evidence, this article aims to overcome the limitations of existing methods and provide more reliable predictions of drug safety risks. The workflow of methodology implementation is shown in Fig. 1.

### Data sources and preprocessing

This study integrates multiple pharmacological and biological data sources to predict drug-drug interactions (DDIs) and their potential adverse effects. The data collection and preprocessing steps are as follows:

(1) FAERS database: We utilize self-reported data from the FDA Adverse Event Reporting System (FAERS), which includes comprehensive reports of adverse events

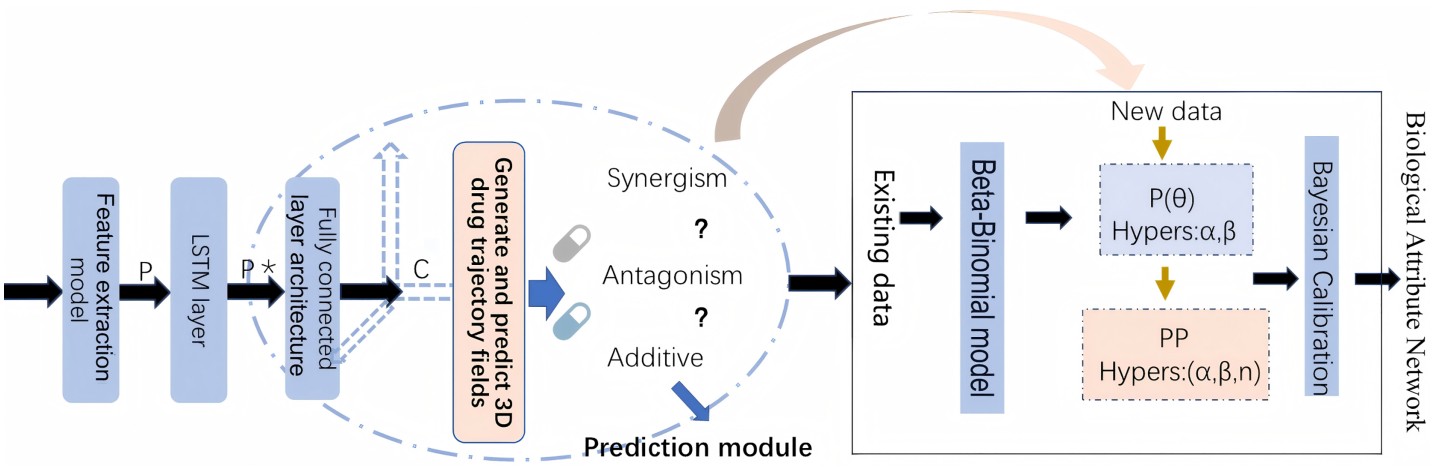

**Figure 1 The workflow of methodology implementation.**

associated with drug combinations (*Yu et al., 2021*). The data spans from March 30, 2021 to March 30, 2023 (2021Q1-2023Q1), providing crucial information on drug usage, adverse events, and patient demographics.These reports provide crucial information on drug usage, adverse events, and patient demographics (*Hou et al., 2016*). The FAERS data is preprocessed to remove duplicate reports, handle missing values, and standardize the formats for drug names and adverse event descriptions. This ensures the consistency and reliability of the dataset used for training.

(2) DrugBank: DrugBank is a publicly available database that contains information on drugs and drug interactions. We have downloaded DrugBank version 5.1.9 in XML format (https://go.drugbank.com/releases/latest#full), and analyze each data tag in the XML file using the lxml parsing library to obtain the corresponding chemical properties such as molecular weight, hydrogen bond count, and atomic composition, as well as pharmacokinetic parameters including bioavailability, half-life, and protein binding rate. Additionally, drug names and their corresponding SMILES representations were parsed into a text file. For example, aspirin (DB00945) exhibits key chemical properties, including a molecular weight of 180.16 g/mol, a hydrogen bond count of 3, and a logP value of 1.31, indicating moderate lipophilicity. It consists of 13 atoms and 13 chemical bonds. Additionally, aspirin is involved in various drug-drug interactions; for instance, it may decrease the excretion rate of Abacavir, potentially leading to higher serum levels. Furthermore, its metabolism can be enhanced when co-administered with Abatacept.

(3) STRING database: STRING is used to integrate protein-protein interaction (PPI) data, offering insights into the biological pathways and molecular functions involved in drug interactions (*Szklarczyk et al., 2021*). We selected only PPIs with a confidence score above 700 (out of 1,000) in humans to ensure data reliability. These datasets provide crucial insights into the molecular mechanisms of drug interactions, enhancing the credibility of our research findings.

**Table 1 Categories of individual drug behavior for control stratification.**

| Category | Symbol | Description | Statistical significance |
|---|---|---|---|
| Concurrent drug AE | D10,1 | Number of adverse events observed during concurrent administration of both drugs | Primary indicator for interaction effects |
| First drug AE | D01,1 | Number of adverse events observed during single administration of the first drug | Baseline comparison for drug 1 |
| Second drug AE | D11,1 | Number of adverse events observed during single administration of the second drug | Baseline comparison for drug 2 |
| Concurrent total | A10,1 | Total number of cases with concurrent administration of both drugs | Denominator for concurrent AE rate |
| First drug total | A01,1 | Total number of cases with single administration of the first drug | Denominator for first drug AE rate |
| Second drug total | A11,1 | Total number of cases with single administration of the second drug | Denominator for second drug AE rate |

(4) Open targets: This resource contributes additional data on drug-disease associations, gene expression, and biological processes, enhancing the biological context and relevance of the DDIs predicted by our model (*Ochoa et al., 2021*). We extract drug-target associations and target-AE associations to assess potential safety risks. Drug names are mapped to RxNorm ingredient-level terms, and targets are identified using Ensembl stable IDs and UniProtKB accession numbers.

The Adverse Event Open Learning through Universal Standardization (AEOLUS) process is used to curate and standardize the publicly available FAERS database, covering reports from Q1 2021 to Q1 2023. Missing values are primarily found in drug names, adverse event (AE) descriptions, and demographic data. Reports lacking drug or AE information are excluded, and ambiguous or incomplete data that could not be standardized are also removed. Drug names are mapped to RxNorm ingredient-level terms, while AE descriptions were standardized to MedDRA preferred terms (PTs) through text normalization, synonym matching, and programmatic mapping. The final curated dataset contained 8,423,659 reports, 41.68% of which involved multiple drugs. In our analysis, all drugs are treated equivalently, regardless of their roles in the reports. The Open Targets and STRING datasets are integrated to support DDI research by providing drug-target associations, target-related adverse events, and high-confidence protein-protein interactions (PPIs).

To systematically analyze the patterns of adverse events in drug interactions, we established a comprehensive categorization system for drug usage scenarios and their corresponding outcomes (Table 1). This classification framework encompasses six distinct categories that capture both individual and concurrent drug administration patterns, along with their associated adverse event occurrences. The categories are defined through three primary dimensions: concurrent usage of drug pairs (D10,1), single drug usage (D01,1 and D11,1), and their respective total sample populations (A10,1, A01,1, and A11,1). This stratification enables precise quantification of adverse event frequencies under different drug administration scenarios, facilitating more accurate assessment of potential

drug-drug interactions. These stratified categories serve as fundamental parameters in our Bayesian framework for detecting and quantifying drug-drug interactions. The ratios between adverse event occurrences (D-series) and their corresponding total populations (A-series) provide crucial information for calculating interaction probabilities and identifying significant drug-drug interactions. In our study, the categories listed in Table 1 serve as key input features for the Bayesian statistical model, providing important statistical information on adverse event (AE) occurrences for both individual and combined drug usage. We use the different drug combination categories in Table 1 to calculate and quantify the adverse event incidence rates for single and combined drug use. These values are integrated as a feature of drug pairs into the deep learning model and used for the final prediction of drug pair interaction types. This helps the model learn different drug interaction patterns, thereby distinguishing between antagonistic and synergistic effects.

All datasets are preprocessed to ensure compatibility across different sources. For instance, drug names are standardized, and biological data such as gene expression and protein interactions are normalized to ensure uniformity. This comprehensive preprocessing allows us to integrate heterogeneous data sources into a unified prediction framework.

## FEATURE EXTRACTION

In this study, we propose an integrated feature extraction framework that synthesizes information from drug molecular structures, biological interactions, genomic data, and temporal patterns to predict DDIs. The entire perspective of the feature extraction process is shown in Fig 2. Our approach incorporates three key feature extraction methods: MFSynDCP (*Dong et al., 2024*), granule-granule interaction (GGI) (*Yu et al., 2024*), and constrained tensor factorization (CTF) (*Han et al., 2024*). Additionally, we integrate LSTM networks to capture temporal dependencies and active learning to enhance the model's ability to focus on the most informative samples, thereby improving the overall prediction accuracy. Moreover, we apply granular learning to screen for key molecular substructures responsible for DDIs. This step, integrated into the GGI feature extraction process, identifies only the relevant substructure interactions, reducing noise and improving the robustness of the Bayesian framework. By focusing on critical substructures, this method enhances the model's sensitivity to significant DDI signals while maintaining biological interpretability.

During the drug molecular feature extraction process, drug compounds are treated as graph structures based on atomic interactions. RDKit is utilized to extract atomic and chemical bond information from the SMILES representation of a drug and construct its corresponding molecular graph. The converted molecular graph represents the overall molecular structure through a series of atoms and atomic bonds, illustrating atomic connectivity and spatial distribution. In these graph-based molecular structures, nodes represent atoms in the drug structure, while edges encode the chemical bond information between atoms.

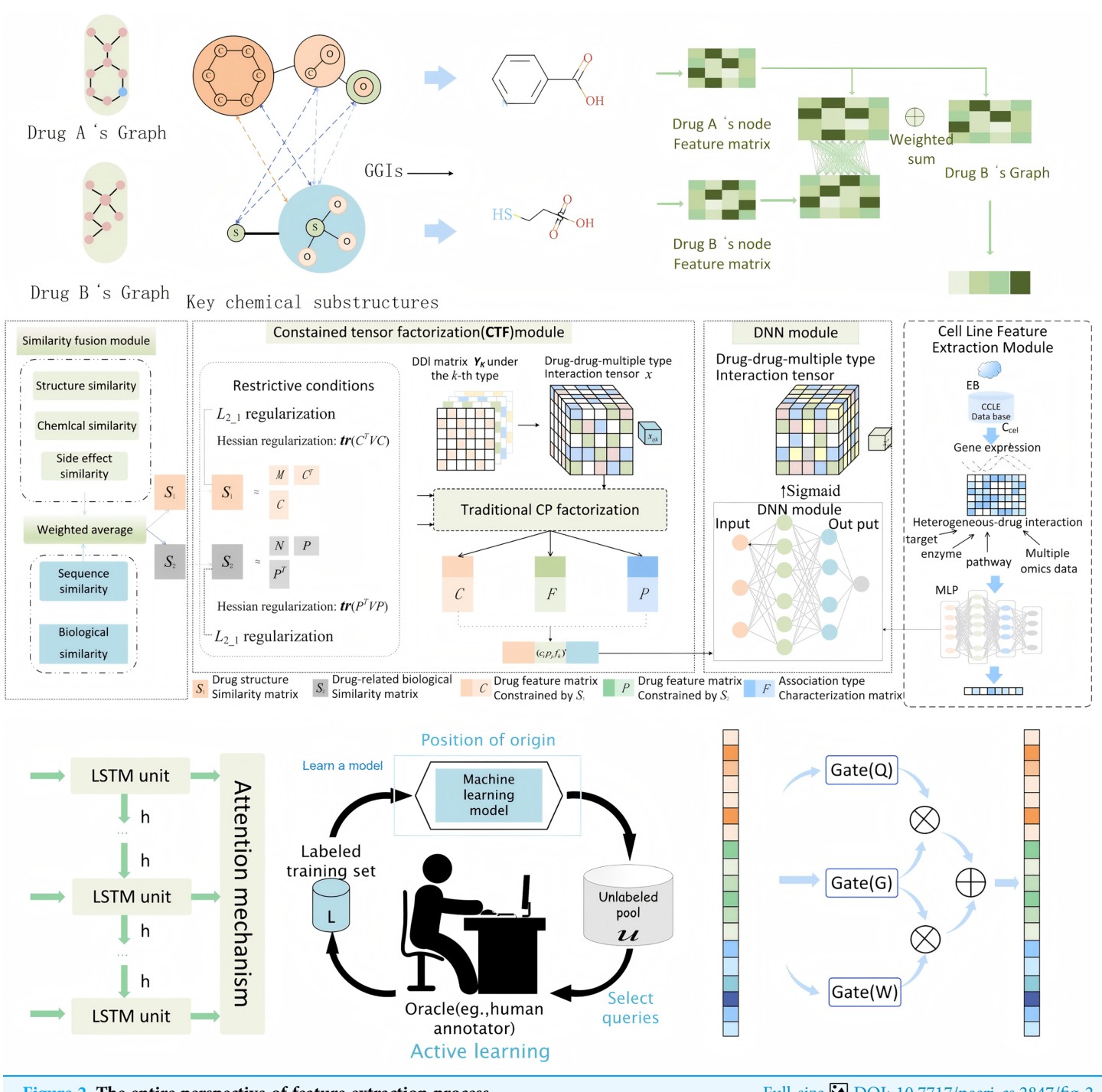

**Figure 2** The entire perspective of feature extraction process.               

# DRUG FEATURE EXTRACTION *VIA* MFSYNDCP

We used an advanced molecular feature extraction framework, MFSynDCP, which demonstrates superior performance in capturing drug molecular characteristics for synergy prediction. The framework begins by converting SMILES strings into molecular

graphs using RDKit, where atoms are represented as nodes and chemical bonds as edges (*Bento et al., 2020*). This fundamental representation preserves the complete molecular topology while enabling sophisticated feature extraction.

In the proposed method, the GAT model is configured with two stacked graph attention layers, each using 16 attention heads, with the multi-head attention results concatenated across layers. In the GAT model, each drug's SMILES representation is converted into a molecular graph using RDKit, where nodes correspond to atoms, and edges denote chemical bonds. The molecular graph of each drug is then encoded by GAT into a high-dimensional semantic feature vector, which is subsequently used for drug interaction type prediction. The model is trained using the gradient backpropagation mechanism to update the parameters. The optimizer employed is Adam, with a learning rate of 1e−4 and weight decay set to 1e−5 to ensure stable parameter updates. During training, the batch size is set to 2,048, and the number of epochs is 200, with early stopping applied (patience = 15 epochs) to prevent overfitting. To enhance the model's generalization ability, regularization techniques are introduced, with a Dropout rate of 0.3 and an EdgeDropout rate of 0.1, further improving the model's robustness and stability. In the graph attention network (GAT) model, the initial format of drugs is a sequence, and each drug's SMILES representation is converted into a molecular graph using RDKit, where nodes correspond to atoms, and edges denote chemical bonds. The molecular graph of each drug is then encoded by GAT into a high-dimensional semantic feature vector, which is subsequently used for drug interaction type prediction.

To effectively extract meaningful features from these molecular graphs, we implement a GAT with an adaptive attention mechanism. In the GAT, for each node i, the attention coefficient $\alpha_{ij}$ with its neighbor j is computed as:

$$\alpha_{ij} = \frac{exp\left(LeakyReLu\left(a^T\left[Wh_i \parallel Wh_j\right]\right)\right)}{\sum_{k \in N_i} exp\left(LeakyReLu\left(a^T\left[Wh_i \parallel Wh_j\right]\right)\right)}$$

where $h_i$ and $h_j$ are node features, W is a learnable weight matrix, a is the attention vector, and II denotes concatenation. The final node representations are then updated through multi-head attention:

$$h_i = \sigma\left[\frac{1}{K}\sum_{k=1}^{k}\sum_{j=N_i} \alpha_{ij}^k W^k h_j\right],$$

where K is the number of attention heads and σ is a nonlinear activation function. The GAT architecture incorporates three key innovations for enhanced molecular structure capture. First, it employs multi-head attention layers that simultaneously learn different aspects of molecular structure, enabling comprehensive feature extraction from multiple chemical perspectives. Second, it implements a hierarchical attention mechanism that progressively aggregates atomic features from local neighborhoods to global structural patterns, effectively capturing both short-range and long-range molecular interactions. Third, it utilizes residual connections between attention layers, ensuring the preservation
of important structural information throughout the feature extraction process. This dual-scale feature extraction proves crucial for model performance, as demonstrated by ablation studies where removal of the MFSynDCP module results in a significant drop in PR_AUC from 0.944 to 0.836.

## GRANULE-GRANULE INTERACTION

To achieve enhanced interpretability in molecular interaction prediction, we incorporate a hierarchical GGI framework that decomposes drug molecules into multiple representational levels. The interaction between two molecular granules $g_i$ and $g_j$ is modeled as: $GGI(g_i, g) = \sigma(W_1 g_i + W_2 g_j + b)$, where $W_1$ and $W_2$ are learnable weight matrices, b is a bias term, and $\sigma$ is a nonlinear activation function. The overall granule interaction score is computed through: $Score_{GGI} = \sum_{i=1}^{N_g} \sum_{j=1}^{N_g} GGI(g_i, g_j) . M_{ij}$, where M is a mask matrix indicating valid granule pairs, and Ng is the number of granules. This approach proves instrumental in capturing complex molecular interactions, as evidenced by ablation studies showing its significant contribution to model performance (removal leads to AUC decrease from 0.947 to 0.862). The GGI framework implements a three-tier molecular representation system: atomic-level features capture fundamental chemical properties, granule-level representations encode functional substructures, and whole-drug features preserve global molecular characteristics. Rather than treating molecules as indivisible entities, this hierarchical approach enables detailed analysis of interaction patterns between specific molecular components. We employ a graph isomorphism network (GINE) to compute inter-granule interactions, with particular emphasis on their functional roles in drug-drug interactions.

## CONSTRAINED TENSOR FACTORIZATION

In addition to structural information, we employ constrained tensor factorization (CTF) to capture similarities between drugs based on both structural and biological data. Our CTF framework constructs a comprehensive similarity tensor incorporating three key dimensions: chemical structure similarity derived from molecular fingerprints, biological interaction profiles extracted from STRING database, and drug-target relationships obtained from DrugBank. The CTF method decomposes this matrix into latent feature representations that capture hidden patterns and relationships between drugs. The CTF optimization problem is formulated as:

$$min_{U,V,W} \ \| \chi - U \circ V \circ W \|_F^2 \ + \lambda_1 \| U \|_{2,1} + \lambda_2 tr(U^T L U),$$

where: $X$ is the original similarity tensor; U, V, W are factor matrices; $\circ$ denotes the outer product, $\| \cdot \| F$ is the Frobenius norm $\| \cdot \| F$; $\| \cdot \| F_{2,1}$ is the $L_{2,1}$ norm for sparsity; $L$ is the Laplacian matrix; $\lambda_1, \lambda_2$ are regularization parameters. By applying Hessian regularization $L_{2,1}$ and regularization, we ensure that the tensor decomposition is robust and focuses on biologically meaningful interactions, leading to more precise feature extraction for downstream prediction tasks.

## FEATURE FUSION WITH CELL LINE FEATURES

Incorporating genomic data from the Cancer Cell Line Encyclopedia (CCLE) allows us to include cell line-specific features, which are crucial for understanding how drug interactions manifest in different biological contexts. We use a multi-layer perceptron (MLP) to process the gene expression data, reducing its dimensionality while preserving essential information. The MLP learns nonlinear relationships within the gene expression profiles, making them compatible with drug feature vectors extracted from previous modules. By using pruning techniques, we remove redundant or irrelevant features, thus optimizing the computational efficiency of the model while ensuring the accuracy of DDI predictions.

We integrate the diverse sources of extracted features—drug molecular structures from MFSynDCP, substructure interactions from GGI, similarity data from CTF, and genomic data from cell lines. The final fused feature vectors, incorporating structural, biological, genomic, and temporal information, are then used for DDI prediction. This robust fusion of multi-modal data allows the model to generate more informed and accurate predictions about drug-drug interactions, improving both prediction reliability and interpretability.

## TEMPORAL FEATURE EXTRACTION *VIA* LSTM

To capture potential temporal dependencies in drug interactions, we integrate a long short-term memory (LSTM) network. After feature extraction modules, the resulting fused feature vectors are passed through the LSTM network. LSTM is particularly effective for modeling sequential data and capturing long-term dependencies, which is useful when drug interactions unfold over time or have latent temporal patterns. By analyzing these temporal sequences, LSTM enhances the model's ability to predict DDIs that evolve or exhibit time-dependent effects, offering a more dynamic and context-aware prediction capability.

## ACTIVE LEARNING FOR SAMPLE SELECTION

To improve the model's training efficiency and focus on the most informative drug-cell interactions, we incorporate an active learning strategy. Active learning allows the model to prioritize samples that carry the highest uncertainty, thereby focusing additional training on these critical interactions. The process begins with the model making initial predictions on a wide range of drug-cell line pairs, and based on the uncertainty in these predictions, it identifies the most uncertain and informative samples. These are then selected for further training, enabling the model to improve its performance while reducing the amount of labeled data required for training. This strategy significantly enhances the model's ability to learn from a diverse yet minimal dataset, optimizing resource use and improving prediction accuracy.

### Prediction model and Bayesian calibration

During model training, we used a subset of the large-scale tumor screening drug combination dataset published by *O'Neil et al. (2016)* as our benchmark dataset. This

dataset involves the screening of 583 different combinations of 22 experimental drugs and 16 approved drugs across 39 cancer cell lines, comprising 23,052 triplets, each consisting of two drugs and a cancer cell line. The dataset primarily includes the SMILES representations of interacting drug pal;irs, a cell line feature, and an interaction classification label. The interaction labels are typically determined based on adverse event types observed in FAERS following drug pair administration. In our model, the SMILES representations of drug pairs and the cell line features serve as inputs, ultimately predicting the type of drug interaction.

In the prediction stage, we integrate multiple techniques to ensure robust and accurate predictions of DDIs. After feature extraction as described in the previous section, the extracted features are input into a multi-layer prediction model that combines recurrent neural networks for sequence learning and a Bayesian framework for model calibration and statistical signal detection.

### DA-RNN for DDI prediction

To account for potential temporal dependencies and interaction sequences between drugs and cell lines, we use a dual-stage attention-based recurrent neural network (DA-RNN). The DA-RNN processes the fused feature vectors derived from the MFSynDCP, GGI, and CTF modules and captures complex sequence dependencies within the data. The dual-stage attention RNN processes the feature sequences through two attention mechanisms:

### Input attention

For each feature k at time t, the attention weight is computed as:

$$e_t^k = V_e^T tanh(W_e[h_{t-1}; s_{t-1}] + U_e X_t^k) \quad \alpha_t^k = \frac{exp(e_t^k)}{\sum_{i=1}^n exp(e_t^i)}$$

where $v_e$, $W_e$, $U_e$ are learnable parameters, $h_{t-1}$ is the previous hidden state, and $s_{t-1}$ is the previous decoder state.

### Temporal attention

The temporal attention weights are computed as:

$$l_t^i = V_t^T tanh(W_t h_i + U_t d_t) \quad \beta_t^i = \frac{exp(l_t^i)}{\sum_{j=1}^T exp(l_t^j)}$$

where $v_t$, $W_t$, $U_t$ are learnable parameters, and $d_t$ is the current decoder state.

By applying attention mechanisms at both feature and sequence levels, the DA-RNN focuses on the most relevant interactions, enhancing the model's ability to predict DDIs. This network allows the model to account for the dynamic nature of drug interactions and potential time-dependent effects in drug responses.

### Bayesian hypothesis testing framework

Following the DA-RNN's predictions of potential DDIs, we employ a "Bayesian Hypothesis Testing Framework" based on the Beta-Binomial model to refine the results for

DDI signal detection $p(\theta) = Beta(\alpha, \beta)$, likelihood function for observed interactions: $p(\mathrm{X}|\theta) = Binomial(n, \theta)$, posterior distribution:

$p(\theta|X) = (\alpha + X, \beta + n - X)$, the Bayes factor (BF) for hypothesis testing is computed as:

$$BF = \frac{P(D|H_1)}{P(D|H_0)} = \frac{\int P(D|\theta)P(\theta|H_1)d\theta}{\int P(D|\theta)P(\theta|H_0)d\theta},$$

where: $H_1$: hypothesis of DDI presence; $H_0$: null hypothesis of no interaction; D: observed data; $p(\theta|H_1)$ and $p(\theta|H_0)$ are prior distributions under each hypothesis. The model classification threshold is determined by: $\tau = argmax_t F1 - score(BF > t)$.

The Beta-Binomial model was selected to address overdispersion in adverse event counts (variance > mean), which is common in sparse pharmacovigilance data. Traditional binomial models assume variance equals mean, but FAERS data exhibit higher variability due to underreporting and heterogeneity. The Beta-Binomial's additional dispersion parameter accommodates this, improving calibration. This framework assesses the statistical significance of predicted interactions by comparing the incidence of adverse events (AEs) between drug combinations and single drugs. Unlike traditional naive Bayes methods, our approach avoids independence assumptions and better captures the joint probability distributions of drug interactions. Through empirical Bayesian methods that optimize prior distributions, the framework effectively handles rare events and underrepresented combinations in the training data, thereby reducing false positives and enhancing the reliability of clinically relevant DDI signal identification.

As for the hyperparameter determination: $\alpha$ and $\beta$: Estimated *via* empirical Bayes using method-of-moments on the observed adverse event rates across all drug pairs. This ensures priors are data-driven rather than arbitrary. Attention Heads in GAT: Eight heads were chosen after grid search (4–12 heads) to balance computational cost and performance. Multi-head attention captures diverse chemical interactions (*e.g.*, functional groups *vs.* ring systems). Learning rate: Initial value of 0.001 with cosine annealing. Determined *via* linear scaling relative to batch size (256), following recommendations for Adam optimizer stability. Batch size (256): Optimized for GPU memory constraints while maintaining gradient estimation accuracy. Early stopping (10 epochs): Monitored validation loss to prevent overfitting, with patience set to avoid premature termination. Optimization: Adam optimizer with $\beta_1 = 0.9$, $\beta_2 = 0.999$. Dropout rate: 0.3, hidden dimensions: 256 for all neural network layers.

### Estimating hyperparameters and Bayesian calibration

Empirical Bayesian methods are used to estimate the hyperparameters of the prior distribution based on existing DDI data. These parameters are progressively optimized during model training, enabling the model to adjust its predictions dynamically and evaluate adverse event rates across different drug combinations. This iterative optimization enhances the model's ability to detect significant signals associated with DDIs. Once the DA-RNN generates the predicted DDI signals, the Bayesian framework is used for calibration. This step helps refine the results by adjusting the signal strength of each

prediction based on the computed posterior probabilities (*Apley, 2012*). By filtering and prioritizing drug combinations with high posterior probabilities, this method reduces false positives and increases the overall prediction accuracy. Hessian and L2,1 regularization are incorporated to minimize overfitting, particularly in scenarios with sparse or imbalanced data.

### Integration of systems pharmacology and validation of biological relevance

To ensure that the predicted DDIs align with known pharmacological mechanisms, we integrate biological data from multiple sources, including DrugBank, STRING, and Open Targets. These databases provide critical information on drug-target interactions, protein-protein interactions, and drug-disease associations, which are used to enrich the biological context of the predictions.

To analyze the relationships between drugs, targets, enzymes, and transporters involved in drug combinations, we construct a biological attribute network. This network links drugs with their corresponding molecular targets, such as enzymes and transport proteins, providing a detailed view of the underlying biological interactions that drive DDIs.

By integrating these biological attributes, we validate the predicted DDIs, ensuring they are not only statistically significant but also biologically plausible. Metrics such as shortest path, enzyme/transporter interactions, and known biological pathways are used to evaluate the relevance of the predictions. This biological validation step enhances the interpretability of the model, providing confidence in the predicted interactions, especially in cases where direct experimental evidence may be lacking.

## EVALUATION METRICS

This section outlines the evaluation metrics used to assess the performance of our proposed model, compares it to other baseline models, details the ablation experiments conducted to evaluate the contributions of specific modules, and evaluates the performance of the Bayesian framework for signal detection. To comprehensively evaluate the predictive performance of our model in detecting DDIs and adverse events, we employed the following key metrics: area under the ROC curve (AUROC), area under the precision-recall curve (PR_AUC), accuracy (ACC), balanced accuracy (BACC), precision (PREC), recall (RECALL)l, F1-score, mean squared error (MSE) and root mean squared error (RMSE).

## EXPERIMENTAL RESULTS

### Comparison experiments

To validate the effectiveness of our proposed model-Dual-stage attention and Bayesian calibration with active learning drug-drug interaction (DABA-DDI), we compare several models on the test dataset. These methods include traditional machine learning models such as random forest (RF), support vector machine (SVM), naïve Bayes (NB), gradient boosting tree (GBT), multilayer perceptron (MLP), and extreme gradient boosting (XGBoost), as well as deep learning models like MGAE-DC and DeepSynergy. These

**Table 2 Comparison of performance metrics between the DABI-DDI model and other models[1].**

| Model | PR_AUC[2] | AUC | BACC | ACC | PREC | MSE | RMSE | F1_Score | RECALL |
|---|---|---|---|---|---|---|---|---|---|
| **DABI-DDI** | **0.944** | **0.947** | **0.879** | **0.879** | **0.876** | **0.094** | **0.307** | **0.880** | **0.884** |
| DeepSynergy | 0.864 | 0.882 | 0.798 | 0.801 | 0.821 | 0.166 | 0.407 | 0.777 | 0.738 |
| MGAE-DC | 0.732 | 0.753 | 0.676 | 0.677 | 0.642 | 0.275 | 0.524 | 0.716 | 0.81 |
| MLP | 0.836 | 0.832 | 0.744 | 0.743 | 0.724 | 0.257 | 0.507 | 0.754 | 0.786 |
| XGBoost | 0.847 | 0.852 | 0.771 | 0.770 | 0.760 | 0.23 | 0.479 | 0.771 | 0.781 |
| RF | 0.696 | 0.731 | 0.676 | 0.676 | 0.672 | 0.324 | 0.569 | 0.672 | 0.671 |
| SVM | 0.766 | 0.779 | 0.710 | 0.710 | 0.720 | 0.29 | 0.538 | 0.697 | 0.675 |
| NB | 0.575 | 0.614 | 0.610 | 0.609 | 0.587 | 0.391 | 0.626 | 0.638 | 0.698 |
| GBT | 0.741 | 0.754 | 0.672 | 0.671 | 0.657 | 0.329 | 0.573 | 0.677 | 0.700 |

Notes:
[1] All metrics are averaged over five runs with different random seeds ($p < 0.05$).
[2] Abbreviations: PR_AUC, Precision-Recall Area Under Curve; AUC, Area Under ROC Curve; BACC, Balanced Accuracy; ACC, Accuracy; PREC, Precision; MSE, Mean Squared Error; RMSE, Root Mean Squared Error.
[3] Bold values indicate best performance.

models' performance metrics are compared with those of the DABI-DDI model in Table 2. In our experiments, the dataset is divided into training and test sets with a 90% to 10% split. We employ five-fold cross-validation on the training set to assess the model's generalization ability. Specifically, the training dataset is randomly divided into five equal subsets, with four subsets used for training and the remaining subset for validation, and the final performance metrics are calculated as the average of the five validation results. This approach effectively reduces the risk of overfitting and provides a more comprehensive evaluation of the model's robustness and performance. Cross-validation maximizes the utilization of training data, allowing the model to encounter more sample data during the training process. To prevent overfitting and ensure generalization, we employ an early stopping mechanism that automatically terminates the training process if validation loss or accuracy does not improve over several iterations. Additionally, we utilize other techniques, such as regularization and learning rate decay strategies, to further enhance the model's robustness and performance.

The methods MDF-SA-DDI, MATT-DDI, and SubGE-DDI were not included in our comparative experiments due to fundamental differences in task scope and data compatibility. MDF-SA-DDI and MATT-DDI are designed for multi-type DDI prediction (*e.g.*, pharmacokinetic *vs.* pharmacodynamic interactions), whereas our work focuses on drug combination synergy prediction with an emphasis on adverse event detection. SubGE-DDI, while innovative in subgraph-enhanced DDI extraction, operates on biomedical text data rather than structured pharmacological datasets like FAERS and DrugBank. Additionally, the lack of publicly available implementations for these models precludes a fair performance comparison. Instead, we selected MGAE-DC and DeepSynergy as baselines due to their alignment with our task (synergy prediction) and reproducible frameworks, ensuring a consistent evaluation protocol.

As shown in Fig. 3A and Table S5, DABI-DDI outperform other methods across multiple key metrics, demonstrating its superior capability in predicting drug combination

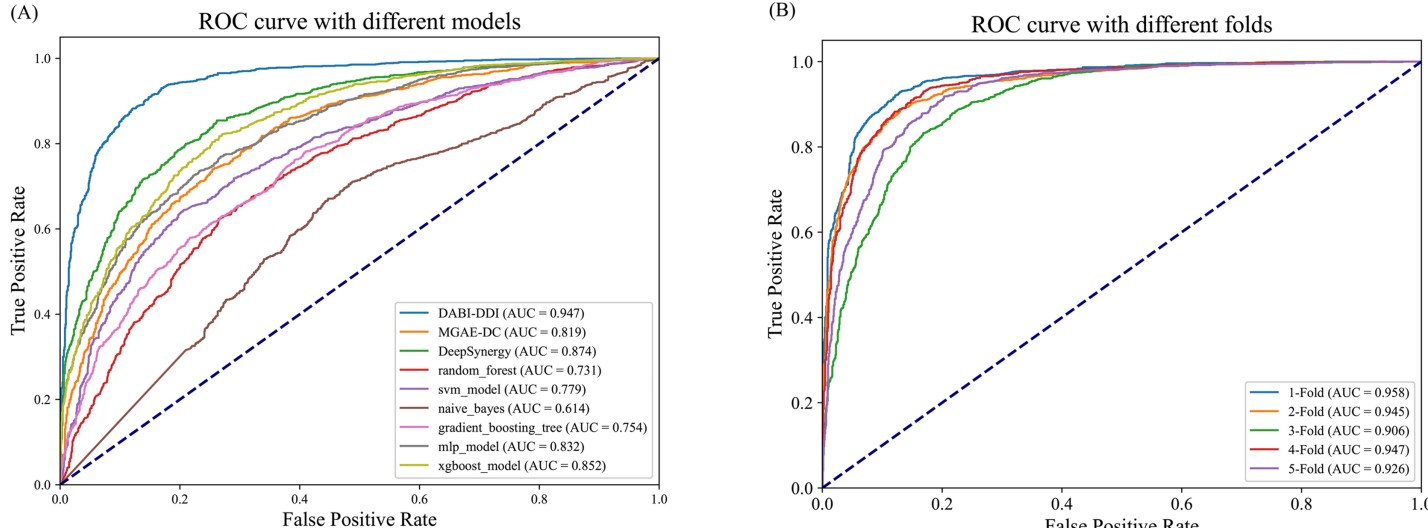

**Figure 3 Performance evaluation of the model.** (A) Comparison of AUC-ROC curves between the DABI-DDI and other models. (B) Comparison of AUC curves from five-fold cross validation.

synergy. The proposed model achieved an AUC of 0.947, surpassing DeepSynergy's 0.882 and other models such as MGAE-DC (AUC of 0.753), XGBoost (AUC of 0.852), and MLP (AUC of 0.832). This indicates that our model has a stronger ability to distinguish between synergistic and non-synergistic drug combinations. Additionally, DABI-DDI obtain a PR_AUC of 0.944, higher than DeepSynergy's 0.864 and XGBoost's 0.847, emphasizing its high precision in identifying synergistic drug combinations while maintaining competitive recall. The overall accuracy (ACC) of DABI-DDI reaches 0.879, exceeding DeepSynergy's 0.798, reflecting its better performance in correctly predicting synergy scores. In terms of balanced accuracy (BACC), our model achievs 0.879, compared to DeepSynergy's 0.798 and XGBoost's 0.771, indicating its ability to handle class imbalance effectively, improving the prediction accuracy for both synergistic and non-synergistic combinations. For precision (PREC), DABI-DDI achieves 0.876, outperforming DeepSynergy's 0.821 and XGBoost's 0.760, which highlights its ability to reduce false positives when predicting synergy. The mean squared error (MSE) of our model is 0.094, significantly better than DeepSynergy's 0.166 and XGBoost's 0.230, demonstrating its higher predictive accuracy. Moreover, DABI-DDI's F1 score and recall are 0.880 and 0.884, respectively, consistently higher than other models, showcasing the robustness and effectiveness of our model in this predictive task.

To validate the effectiveness of the model in predicting drug combination synergy, we design a five-fold cross-validation experiment on the training set to assess the performance of DABI-DDI across different folds. The dataset is divided into five folds, with each fold used for both training and testing. In each fold, the model is trained on four folds and validated on the remaining one. The experimental results are shown in Table S5, while Fig. 3B presents a comparison of AUC curves across the different folds. As indicated in

Table S5, the PR_AUC and AUC values range from 0.914 to 0.958 across the folds, demonstrating the model's high accuracy in predicting drug combination synergy. Specifically, the PR_AUC for the 1-Fold reaches 0.954, while the AUC for the 1-Fold is 0.958, representing the best performance. Both BACC and ACC also show good results, maintaining values between 0.856 and 0.898, which reflects the model's balanced and accurate classification of positive and negative samples. The PREC and RECALL values further highlight the model's reliability in identifying synergistic drug combinations. In particular, the 1-Fold achieves a PREC of 0.889 and a RECALL of 0.905, indicating that the model maintain a high recall rate while improving precision. The MSE and RMSE results also suggest low prediction errors, with MSE values fluctuating between 0.08 and 0.29, illustrating the model's stability in predicting drug combination synergy. From Table S5, we can observe that as the false positive rate increases, the true positive rate nearly reaches 1.0, indicating the model's strong discriminative ability in distinguishing between synergistic and non-synergistic drug combinations. The AUC value of 0.958 for the two-Fold, the highest among all folds, further confirms the model's superior performance.

## ABLATION EXPERIMENTS

### Evaluation of different modules

To verify the effectiveness of each module in the proposed model, we conducted ablation experiments by comparing the performance of different module combinations shown in Fig. 4 and Fig. S1. The full model, DABI-DDI, includes all components—MFSynDCP, GGI, CTF, LSTM, active learning, and Bayesian correction. The following variations were tested: No MFSynDCP: This setting removes the MFSynDCP module to assess the importance of GAT-based molecular structure extraction. No GGI: The GGI module is excluded to evaluate the impact of granule-level interactions on model performance. No CTF: We eliminate the CTF module to test the significance of drug similarity information for predicting DDIs. No LSTM: The LSTM module is removed to evaluate the contribution of temporal dependencies to prediction accuracy. No active learning: Active learning is omitted to measure its role in enhancing model efficiency and accuracy. No Bayesian correction: The Bayesian correction is disabled to assess its influence on signal correction, false positives, and overall model reliability. The experimental results are presented in Table S6. To better observe the performance of different module combinations across various metrics, Table S7 illustrates the variations in these metrics for the different module combinations. As shown in Table S6, when removing the MFSynDCP module, we observe a drop in PR_AUC from 0.944 to 0.836 and AUC from 0.947 to 0.845, suggesting that the MFSynDCP module plays a crucial role in capturing the structural characteristics of drug interactions. The removal of GGI leads to a significant drop in performance across almost all metrics, with the PR_AUC decreasing to 0.864 and AUC to 0.862. This suggests that the GGI module, which models molecular granule-level interactions, contributes significantly to the overall accuracy, with ACC declining from 0.879 to 0.771. The relatively stable recall value of 0.833 compared to other ablations indicates that GGI is particularly crucial for improving the precision and specificity of the model, but less so for identifying true

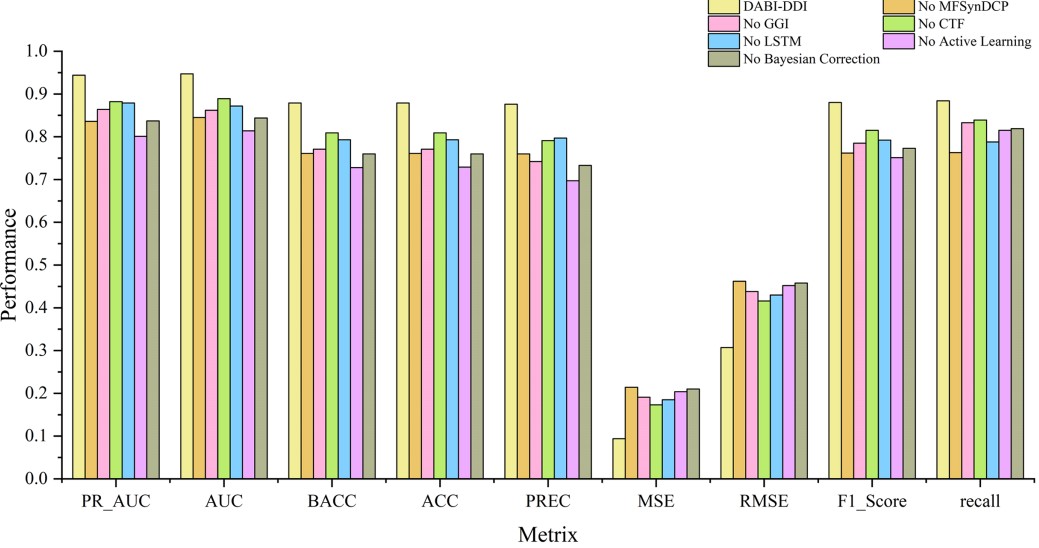

**Figure 4 Comparison of metrics for different module combinations.**

positives. Similarly, the removal of CTF results in performance degradation across metrics, with PR_AUC and AUC dropping to 0.882 and 0.889, respectively. When the Bayesian correction is removed, the performance drop is the most pronounced, with PR_AUC plunging to 0.837 and AUC to 0.844, clearly indicating that the Bayesian correction is key to enhancing the model's prediction confidence and accuracy. The MSE increases dramatically to 0.210, and the F1 score falls to 0.773, confirming that the Bayesian framework significantly improves the model's precision and error reduction.

To better analyze the contribution of each module, we conducted dual ablation experiments to investigate their synergistic effects. The experimental results are shown in Table S8. Additionally, the heatmaps after removing different pairs of substructures are shown in Fig. S2, where (a) shows the heatmap without MFSynDCP and GGI structures, (b) shows the heatmap wiuthout MFSynDCP and CTF structures, (c) shows the heatmap without MFSynDCP and LSTM strctures, (d) shows the heatmap without GGI and CTF structures, and (e) shows the heatmap without GGI and LSTM structures. The results in Table S8 showes that the joint removal of MFSynDCP and GGI led to a 30% decrease in AUC (from 0.947 to 0.665), which is significantly greater than the effect of removing MFSynDCP alone (10% decrease) or GGI alone (8% decrease). This indicates a strong synergy between structural feature extraction (MFSynDCP) and molecular-level interaction modeling (GGI). Similarly, the joint removal of MFSynDCP and CTF resulted in a 27% decrease in AUC (from 0.947 to 0.694), which exceeded the individual effects of removing MFSynDCP (10% decrease) and CTF (5.8% decrease). This suggests that structural drug representation (MFSynDCP) and similarity modeling (CTF) provide complementary information for drug interaction prediction.Furthermore, removing both CTF and LSTM together led to a 16% drop in AUC (from 0.947 to 0.793), whereas removing CTF alone caused a 5.8% decrease and removing LSTM alone resulted in a 7.5%

decrease. This highlights the complementary roles of CTF in similarity modeling and LSTM in capturing temporal dependencies.

Additionally, the joint removal of GGI and LSTM caused an 18% decline in AUC (from 0.947 to 0.764), which was greater than the effects of removing GGI alone (8.5% decrease) or LSTM alone (7.5% decrease). This demonstrates a significant synergy between molecular granularity features (GGI) and temporal learning (LSTM). These results indicate that feature fusion across modules plays a crucial role in drug interaction prediction, and the complementary and synergistic effects among different modules significantly enhance the model's predictive accuracy.

## EFFICIENCY EVALUATION EXPERIMENTS OF BAYES MODEL

To evaluate the performance of drug combinations regarding the probability of adverse events, we employ a Bayesian beta-binomial model to calculate the posterior distribution of adverse event probabilities for combination drugs and single drugs using the FAERS adverse event dataset. We estimate the hyperparameters $\partial$ and $\beta$ of the prior distribution based on the adverse event occurrence rate of combination drugs using an empirical Bayesian approach. By calculating the sample mean and variance of the occurrence rates for combination drugs, we derive the prior parameters. Using these prior parameters, the model effectively combined prior knowledge with observational data to output updated posterior probabilities. The experimental results are illustrated in Figs. 5A, 5B. The KDE illustrates the posterior probability distributions for combination drugs (in green) and single drugs (in orange and blue). The density curve for combination drugs resembles that of single drugs in the lower posterior probability range but extends into higher probability values, indicating that certain combination drugs carry a greater risk of adverse events. In contrast, the posterior probability distribution for single drugs is more concentrated in the low probability interval. The violin plot further depicts the distribution shapes of posterior probabilities for both combination and single drugs. The distribution for combination drugs is broader, showing a wider spread near zero, which suggests that while most combination drugs have lower posterior probabilities, some approach a probability of 1.0. Conversely, the posterior probabilities for single drugs are primarily concentrated in a lower range, indicating that the likelihood of adverse events for single drugs is generally low. This analysis suggests that the risk of experiencing adverse events is higher with combination drugs compared to single drugs, particularly for specific drug combinations that show significantly increased risks. This heightened risk may be attributed to the synergistic effects of these combinations, which make them more likely to induce adverse events than individual drugs.

To evaluate the performance of the Bayesian model in predicting adverse events, we compare it with several signal detection algorithms, including Omega, Log-likelihood Ratio (LLR), Interaction Signal Score (IntSS), and delta_add. The results are shown in Fig. 5C, according to the experimental results, the Bayesian model achieved a significantly higher AUC value (0.94) compared to the other signal detection algorithms, such as Omega (AUC = 0.54), LLR (AUC = 0.52), and IntSS (AUC = 0.48). This demonstrates that

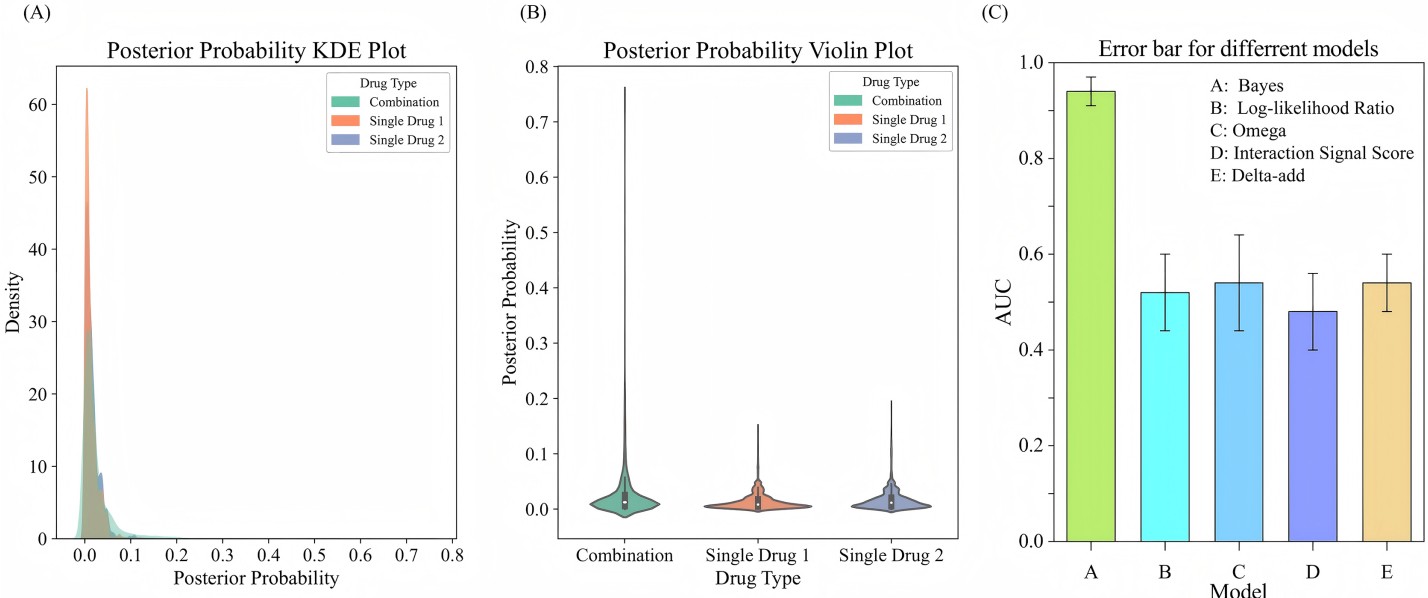

**Figure 5** **Comparison of posterior probability distribution and model performance in predicting adverse events.** (A) KDE plot of posterior probability distribution for adverse events comparing drug combinations with single drugs. (B) Violin plot showing the distribution of posterior probabilities for drug combinations *vs.* single drugs. (C) Error bars for comparing different models for adverse event prediction.

the Bayesian model offers superior accuracy in predicting adverse events. The improved performance of the Bayesian model is due to its ability to effectively utilize posterior probabilities in combination with drug combination features, allowing for more precise identification of adverse event risks. In contrast, signal detection algorithms typically rely on pre-established statistical metrics to score the interactions of drug combinations, and their performance in classification tasks may be limited by inherent constraints in their algorithms. By incorporating posterior probabilities and usage frequency as features, the Bayesian classifier outperformed traditional signal detection algorithms in predicting adverse events. This result highlights the efficacy of our model in predicting the risks associated with adverse events from drug combinations, offering a more reliable approach than conventional methods.

The posterior probabilities inferred using the Bayesian model often encapsulate information about the synergistic or antagonistic effects of drug combinations. As such, these probabilities can serve as features to predict whether drug interactions exhibit synergy or antagonism. To verify the effectiveness of using posterior probability as a feature, we incorporate the usage frequency of different drug combinations as features and used posterior probabilities as additional feature columns. A Bayesian classifier is then trained to predict the category of adverse events. For the same adverse event, the model's confidence level across various drug combinations is ranked. The ranking enables the identification of the most likely drug combinations associated with a specific adverse event. The relevant experimental results are shown in Fig. 6. We analyzed four types of adverse

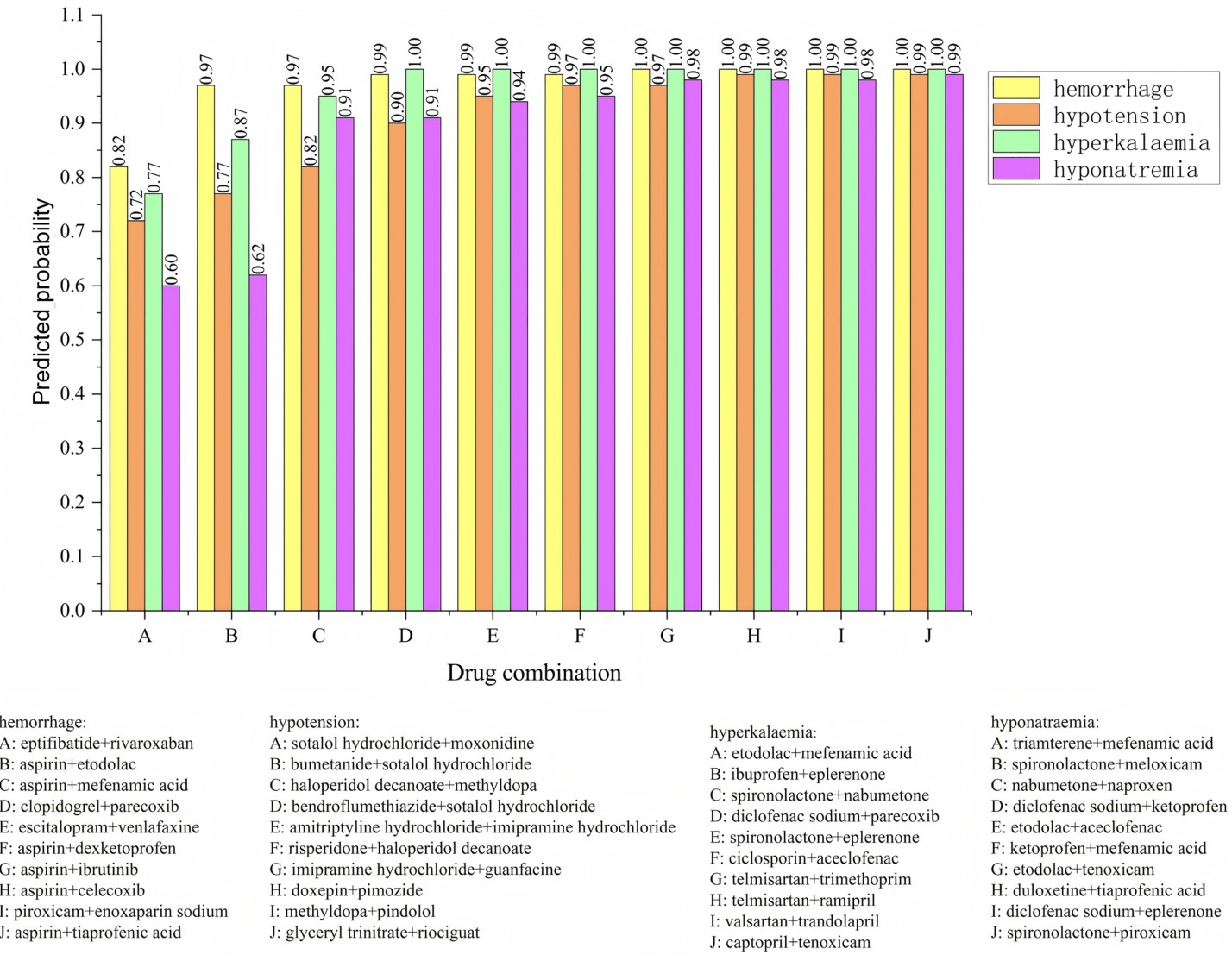

hemorrhage:
A: eptifibatide+rivaroxaban
B: aspirin+etodolac
C: aspirin+mefenamic acid
D: clopidogrel+parecoxib
E: escitalopram+venlafaxine
F: aspirin+dexketoprofen
G: aspirin+ibrutinib
H: aspirin+celecoxib
I: piroxicam+enoxaparin sodium
J: aspirin+tiaprofenic acid

hypotension:
A: sotalol hydrochloride+moxonidine
B: bumetanide+sotalol hydrochloride
C: haloperidol decanoate+methyldopa
D: bendroflumethiazide+sotalol hydrochloride
E: amitriptyline hydrochloride+imipramine hydrochloride
F: risperidone+haloperidol decanoate
G: imipramine hydrochloride+guanfacine
H: doxepin+pimozide
I: methyldopa+pindolol
J: glyceryl trinitrate+riociguat

hyperkalaemia:
A: etodolac+mefenamic acid
B: ibuprofen+eplerenone
C: spironolactone+nabumetone
D: diclofenac sodium+parecoxib
E: spironolactone+eplerenone
F: ciclosporin+aceclofenac
G: telmisartan+trimethoprim
H: telmisartan+ramipril
I: valsartan+trandolapril
J: captopril+tenoxicam

hyponatraemia:
A: triamterene+mefenamic acid
B: spironolactone+meloxicam
C: nabumetone+naproxen
D: diclofenac sodium+ketoprofen
E: etodolac+aceclofenac
F: ketoprofen+mefenamic acid
G: etodolac+tenoxicam
H: duloxetine+tiaprofenic acid
I: diclofenac sodium+eplerenone
J: spironolactone+piroxicam

**Figure 6 Confidence ranking for different drug combinations under the same adverse event using the Bayesian model.**

events: (a) hemorrhage, (b) hypotension, (c) hyperkalaemia (discontinuation), and (d) hyponatremia (induced by administration without fluid restriction). Drug combinations A, B, C, D, E, F, G, H, I, and J represent different combinations. For the adverse event hypotensios, it is evident from the figure that the model's confidence scores varied significantly across different drug combinations. The top five ranked combinations are as follows: sotalol hydrochloride + moxonidine (score = 0.72), sotalol hydrochloride + bumetanide (score = 0.77), haloperidol decanoate + methyldopa (score = 0.82), sotalol hydrochloride + bendroflumethiazide (score = 0.90), and amitriptyline hydrochloride + imipramine hydrochloride (score = 0.95). In the case of the adverse event hypotension, the model's confidence scores reflec the likelihood of each drug combination causing the event.

Higher confidence scores indicat a greater probability that the combination would lead to hypotension. For example, the high confidence scores for sotalol hydrochloride + moxonidine and sotalol hydrochloride + bumetanide can be attributed to their synergistic effects on the blood pressure regulation system. Sotalol, a β-blocker, lowers blood pressure, while moxonidine, a selective I1 receptor agonist, further enhances the hypotensive effect. Similarly, the combination of haloperidol decanoate and methyldopa had a high confidence score, likely due to their dual impact on the central nervous system and blood pressure regulation.

Methyldopa reduces blood pressure by decreasing peripheral resistance, while haloperidol, an antipsychotic, may indirectly affect blood pressure regulation. These confidence distributions align with the known pharmacological actions of the drug combinations, indicating that the model's predictions are both credible and effective. The results reasonably reflect the potential risks of drug combinations in causing adverse events such as hypotension.

In this research, we develop a 'Bayesian classifier' to predict the likelihood of adverse events caused by drug combinations, using the usage frequency of the combinations and their posterior probabilities as features for classifying adverse event categories.

## IMPORTANCE ANALYSIS OF CHEMICAL SUBSTRUCTURES

This study presents a deep learning model named DABI-DDI, which is designed to enhance the accuracy of drug combination predictions while ensuring the interpretability of the results, thereby overcoming the black-box nature of traditional models. The model employs a message-passing mechanism, which improves feature representation between nodes. Additionally, an adaptive attention mechanism is introduced, based on graph aggregation, to score the importance of key chemical substructures in drugs. Experimental results indicate that as the model trains, the attention scores gradually concentrate on specific critical structures, helping to reveal the essential chemical properties that influence drug synergy. Figure S3A displays the visualization results for four drug combinations in the NCIH1650 non-small cell lung cancer cell line, highlighting the role of key chemical substructures in drug synergy. NCIH1650 is a widely used non-small cell lung cancer cell line in cancer drug research. As shown in Fig. S3A, four different drug combinations are provided. In the combination of Etoposide and Dasatinib, the model identifies aromatic rings and amide groups, indicating that these structures play a critical role in signal transduction inhibition and drug-target interactions. For the combination of MK-4827 and MK-2206, the model marks the nitrogen-containing heterocyclic structure as an important substructure, which facilitates drug binding to enzymes and enhances apoptosis. The combination of SN-38 and Sorafenib reveals aromatic structures containing heterocycles, potentially increasing the cytotoxicity of the drugs and inhibiting cancer cell proliferation. Finally, in the combination of AZD1775 and Dasatinib, the model emphasizes the aromatic rings and amide groups, which play significant roles in kinase inhibition and cell cycle blockage. The model's successful identification of these key substructures validates its effectiveness and interpretability in predicting drug synergy.

## Biological analysis in drug combination

In the process of analyzing the interaction between drug combinations in cell lines, factors such as targets, transporters, and enzymes are often involved. To investigate the relationships among these factors during the action of drug combinations, we established an attribute network to examine the factors promoting drug synergy. In this experiment, we selected drug combinations relevant to the NCIH1650 cell line. The drug-target-enzyme-transporter interactions in Fig. S3B were identified using a multi-step approach: Data Integration: Interactions were extracted from DrugBank (for drug-target relationships), STRING (for protein-protein interactions), and Open Targets (for biological pathway context). Statistical filtering: Only interactions with a confidence score >0.7 (STRING) or experimental validation (DrugBank) were retained. For enzymes and transporters, associations were prioritized if they appeared in ≥3 independent studies (PubMed/MEDLINE) or were annotated as "highly relevant" in DrugBank. Network pharmacology: A shortest-path algorithm was applied to connect drugs to adverse events (AEs) *via* shared enzymes/transporters, ensuring biological plausibility. Edge weights were calculated using interaction frequency: The number of literature reports supporting the interaction (normalized to 0–1). Functional relevance: Enzyme/transporter roles in drug metabolism (*e.g.*, CYP3A4 for metabolism) or distribution (*e.g.*, ABCB1 for efflux), weighted by their known involvement in the AE (*e.g.*, CYP2D6 inhibition linked to hypotension). Computational scores: Bayesian posterior probabilities from the model, scaled to reflect interaction strength.

Figure S3B (a) illustrates the synergistic effect between the drugs Doxorubicin and Dasatinib, while Fig. S3B (b) demonstrates the synergy between Etoposide and Dasatinib. NCIH1650, a cell line commonly used in lung cancer research, serves as the experimental model. Doxorubicin, a traditional chemotherapy drug, primarily acts through interactions with the Cytochrome P450 enzyme, inhibiting the proliferation of cancer cells. From Fig. S3B, we can observe that Doxorubicin and Dasatinib are connected through Cytochrome P450 enzymes and transport proteins such as ABCB1 and ABCG2, which play critical roles in their interaction. Similarly, in the interaction between Etoposide and Dasatinib shown in Fig. S3B (b), Cytochrome P450 enzymes and transport proteins are key factors. However, it is evident that the influence of enzymes is more significant than that of transport proteins in the synergy between Etoposide and Dasatinib.

To enhance the practical utility of our work, we propose adding detailed case studies for high-confidence drug pairs identified in Fig. 6. Table S9 shows three illustrative examples demonstrating the clinical relevance and biological plausibility of the model's predictions.

## DISCUSSION

This study presents a novel integration of Bayesian methods with deep learning for DDI prediction, offering significant improvements in both prediction accuracy and biological interpretability. While previous studies have explored various machine learning approaches for DDI prediction, and Bayesian methods have been applied in other areas of drug discovery, our work represents the first comprehensive attempt to combine Bayesian calibration with deep learning specifically for DDI prediction.

To be specific, this study introduces two key innovations: a "Bayesian Calibration" method based on the "Bayesian Hypothesis Testing Framework and Biological Analysis" in drug combinations. The "Bayesian Calibration" plays a pivotal role in refining the model's predictions by incorporating prior knowledge and probabilistic correction mechanisms. As demonstrated in the ablation results, removing this component leads to a marked increase in error rates, with MSE rising from 0.094 to 0.21 and rmse from 0.307 to 0.458, which underscores its importance in improving prediction reliability. This approach enables the model to adjust for uncertainty and systematically correct biases, thereby enhancing overall accuracy and robustness.

Recent works in DDI prediction have primarily focused on improving prediction accuracy through deep learning architectures. For instance, *Deng et al. (2020)* proposed a graph neural network-based model (MGNN) for DDI prediction achieving high accuracy but lacking statistical validation. Similarly, *Karim et al. (2019)* developed a deep learning framework focusing solely on performance metrics. However, these approaches often struggle with false positives and limited biological interpretability, which are crucial for clinical applications.

We have not yet abandoned efforts to enhance the accuracy of the model. In our experiments, we compare traditional machine learning-based methods and deep learning-based approaches, including models such as random forest (RF), support vector machine (SVM), naïve Bayes (NB), gradient boosting tree (GBT), multilayer perceptron (MLP), and extreme gradient boosting (XGBoost). Specifically, the GBT model, known for its ability to sequentially build decision trees while focusing on correcting errors from previous iterations, performs well in capturing complex non-linear relationships in the data. However, its performance can be sensitive to hyperparameter tuning and may require significant computational resources. On the other hand, XGBoost, an optimized version of GBT, is particularly effective in handling sparse data and mitigating overfitting due to its implementation of regularization techniques. Moreover, DeepSynergy integrates both chemical and genomic data to predict drug synergy by employing deep learning models. By using normalization strategies to address the heterogeneity of these datasets, it enhances the model's capacity to capture the intricate relationships between drugs and cell lines. Meanwhile, MGAE-DC incorporates an attention mechanism to fuse drug embeddings across different cell lines, allowing for the extraction of unified drug representations. This attention mechanism helps to prioritize the most relevant information for drug interaction prediction, further refining the model's capabilities. The results in Table S5 show that our proposed model achieves the best results across all metrics, with PR_AUC, AUC, BACC, ACC, PREC, MSE, RMSE, F1_Score, and RECALL reaching 0.944, 0.947, 0.879, 0.879, 0.876, 0.094, 0.307, 0.880, and 0.884, respectively. Furthermore, as seen in Fig. S3A, which illustrates the distribution of importance scores for drug substructures, our model accurately identifies key chemical substructures, thus enhancing the interpretability of the final predictions. This capability not only allows the model to predict drug interactions effectively but also provides valuable insights into the underlying chemical components driving these interactions. *Zhong et al. (2024)* proposed a knowledge graph-based method designed for predicting multi-typed DDIs using contrastive learning. The approach

captures both comprehensive drug-related information and the structural relationships between drugs. While their method refines feature extraction by leveraging knowledge and topological aggregation, the overall complexity of the model may require extensive computational resources, limiting scalability. *Zhu et al. (2022)* offers a significant advancement in adverse drug-drug interaction (ADDI) prediction by employing joint feature selection and a GAN framework, which enhances accuracy and effectively integrates multi-attribute information. However, the model's complexity and its dependency on high-quality data introduce potential limitations for scalability and generalizability in real-world applications.

As shown in Tables S6–S8, the results clearly demonstrate the effectiveness of the proposed model, with each component contributing significantly to overall performance. The full model consistently outperforms its ablated versions across all metrics, underscoring the importance of each module. Notably, the substantial drop in PR_AUC (from 0.944 to 0.836) and AUC (from 0.947 to 0.845) when the MFSynDCP module is removed highlights its critical role in capturing molecular features necessary for accurate predictions. Despite the varied impact of each module, the full model integrates structural, temporal, and uncertainty-based features more effectively, achieving superior predictive performance across all tasks. These findings validate the proposed model's architecture and the synergistic effect of its components in enhancing prediction accuracy and robustness for drug-drug interaction tasks.

The integration of our Bayesian calibration method demonstrates substantial improvements in model reliability. This improvement aligns with findings from other domains where Bayesian methods have enhanced prediction reliability (*Borgia et al., 2021*). Our approach's ability to incorporate prior knowledge and uncertainty quantification addresses a significant gap in existing DDI prediction methods, as highlighted by *Kuksa et al. (2020)* in their review of current challenges in DDI prediction.

The biological network in Fig. S3B elucidates how shared metabolic enzymes and transporters drive DDIs. For instance, the prominence of CYP3A4 and ABCB1 in Doxorubicin-Dasatinib interactions underscores the importance of pharmacokinetic overlap in DDI risk (*Zhan et al., 2020*). Similarly, Etoposide-Dasatinib synergy *via* ABCG2 highlights transporters as modulators of both efficacy and toxicity (*Ait-Oudhia, Ovacik & Mager, 2017*). These patterns align with the "dual exposure" hypothesis, where concurrent drugs compete for shared pathways, exacerbating adverse outcomes. By prioritizing interactions with high functional relevance and literature support, our model bridges computational predictions to mechanistic pharmacology, offering clinicians actionable insights into DDI risks.

The practical implications of our approach are noteworthy for clinical applications. By reducing false positive rates and improving interpretability, our model shows significant potential for drug safety assessment in clinical trials and personalized medicine applications. Furthermore, the framework's ability to provide mechanistic insights makes it valuable for drug repurposing initiatives and clinical decision support systems,

potentially streamlining the drug development process and enhancing patient care. Furthermore, the incorporation of Biological Analysis significantly strengthens the model's ability to understand drug combinations at a deeper level by integrating biological interaction data such as protein-protein interactions and ligand-based similarities. This biological perspective allows the model to capture more biologically meaningful patterns, as reflected in the high AUC and F1 score values achieved by the full model. This biologically-informed approach ensures that the predictions align more closely with real-world drug mechanisms, making the model not only a powerful predictive tool but also a valuable asset for guiding biological interpretations of drug combinations. These advantages position our model as a cutting-edge solution in drug interaction prediction, combining structural insights, probabilistic calibration, and biological relevance to achieve state-of-the-art performance.

Despite the promising results, our study has several limitations that warrant further investigation. The computational overhead introduced by the Bayesian calibration process poses challenges for real-time applications, necessitating the development of more efficient approximation methods. Additionally, while our model demonstrates robust performance on available data, the prediction of rare drug combinations remains challenging due to limited training examples, a common issue in DDI studies (*Ryu, Kim & Lee, 2018*). Although we incorporate biological analysis into our framework, additional experimental validation would further strengthen the biological interpretability of our predictions.

Looking ahead, several promising research directions emerge. Future work should focus on integrating molecular structure information to enhance mechanistic understanding and developing more computationally efficient Bayesian methods. The extension of our framework to handle multi-drug interactions beyond pair-wise predictions and the incorporation of temporal drug administration patterns represent important next steps in advancing this field.

## CONCLUSION

This work represents a significant step toward more accurate and interpretable DDI prediction, offering practical tools for drug safety assessment and combination therapy design in clinical practice. First, we innovatively combine multi-source feature extraction techniques to capture comprehensive drug characteristics from molecular structures to biological interactions. Second, we propose a dual-stage attention mechanism integrated with Bayesian calibration, which significantly enhances the model's ability to detect DDI signals while maintaining biological relevance. The incorporation of active learning strategies optimizes sample selection, making the model more efficient in handling large-scale drug combination data. Third, our framework successfully bridges the gap between statistical signal detection and biological mechanism understanding through the integration of systems pharmacology approaches. Looking forward, these advances take a significant step forward in advancing precision medicine in healthcare, furthering the development of personalized therapeutic strategies.

### Funding

The authors received no funding for this work.

### Competing Interests

The authors declare that they have no competing interests.

### Author Contributions

- Rongpei Li conceived and designed the experiments, performed the experiments, analyzed the data, performed the computation work, prepared figures and/or tables, authored or reviewed drafts of the article, and approved the final draft.
- Yufang Zhang conceived and designed the experiments, prepared figures and/or tables, and approved the final draft.
- Heqi Sun conceived and designed the experiments, prepared figures and/or tables, and approved the final draft.
- Shenggeng Lin conceived and designed the experiments, prepared figures and/or tables, and approved the final draft.
- Guihua Jia conceived and designed the experiments, prepared figures and/or tables, and approved the final draft.
- Yitian Fang conceived and designed the experiments, prepared figures and/or tables, and approved the final draft.
- Chen Zhang conceived and designed the experiments, prepared figures and/or tables, and approved the final draft.
- Xiaotong Song conceived and designed the experiments, prepared figures and/or tables, and approved the final draft.
- Jianwei Zhao conceived and designed the experiments, prepared figures and/or tables, and approved the final draft.
- Lyubin Hu conceived and designed the experiments, prepared figures and/or tables, and approved the final draft.
- Yajing Yuan conceived and designed the experiments, prepared figures and/or tables, and approved the final draft.
- Xueying Mao conceived and designed the experiments, prepared figures and/or tables, and approved the final draft.
- Jiayi Li conceived and designed the experiments, prepared figures and/or tables, and approved the final draft.
- Aman Kaushik conceived and designed the experiments, prepared figures and/or tables, and approved the final draft.
- Dandan An conceived and designed the experiments, prepared figures and/or tables, and approved the final draft.
- Dongqing Wei conceived and designed the experiments, authored or reviewed drafts of the article, guidance of thought, and approved the final draft.

## Data Availability

The data is available at Zenodo: Li, R. (2025). DABI-DDI [Data set]. Zenodo. https://zenodo.org/records/15063733.

The third party datasets are available at:

- https://fis.fda.gov/extensions/FPD-QDE-FAERS/FPD-QDE-FAERS.html
- https://go.drugbank.com/releases/latest.
- https://string-db.org/cgi/download.pl.
- https://platform.opentargets.org/downloads.

## Supplemental Information

Supplemental information for this article can be found online at http://dx.doi.org/10.7717/peerj-cs.2847#supplemental-information.

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
