# Peer review of "Towards interpretable drug interaction prediction via dual-stage attention and Bayesian calibration with active learning"

_PeerJ Computer Science, doi:10.7717/peerj-cs.2847_

## Round 0.1 · original submission · Major Revisions

Please carefully review the feedback and revise the manuscript accordingly. It is essential to provide the curated dataset and the complete code, along with clear, step-by-step instructions on how to reproduce the results. This is a critical requirement for ensuring the transparency, reproducibility, and credibility of your study.

Reviewer 1 ·

Basic reporting

Regarding the structure of the paper, the existing deep learning-based methods mentioned from line 97 do not clearly state their drawbacks, so it is unclear what the problems are and how the proposed method addresses them.
Additionally, in the experimental section, the comparison methods MGAE-DC and DeepSynergy are chosen, I think these should be mentioned in the introduction, and should be properly cited.

The methods MDF-SA-DDI, MATT-DDI, and SubGE-DDI introduced in the introduction, are not compared in the experimental section. The paper should explain the reason for this lack of comparison.

Experimental design

Regarding Table 4, instead of focusing on which folds are easy or hard, it would be better to demonstrate the proposed method's effectiveness on each fold through comparisons with other methods or ablation studies.

Validity of the findings

no comment

Reviewer 2 ·

Basic reporting

The manuscript is generally well-written with clear professional English throughout. The manuscript structure is logical and presents the methodological framework and results in a systematic manner. The mathematical notations show some inconsistency, particularly in the Bayesian framework section. For example, in line 363, the formula "τ = \argmax_t F1-score(BF > t)" needs proper mathematical formatting, and similar issues exist throughout the paper. The literature review is comprehensive and up-to-date, providing good coverage of both traditional methods and recent deep learning approaches in DDI prediction.
The figures and tables effectively present the experimental results and model architecture. Tables 2-4 clearly show performance metrics and comparisons across different methods. Figure 7 successfully illustrates both chemical substructures and biological networks. The data sources (FAERS, DrugBank, STRING databases) are appropriately cited and their usage is well documented, though more details on data preprocessing would be beneficial.

Experimental design

The research presents an original and innovative integration of dual-stage attention mechanisms with Bayesian calibration for DDI prediction. The motivation is clearly established through DDI-related adverse events statistics from FAERS data. The knowledge gap in interpretable DDI prediction is well articulated, particularly regarding the need for improved accuracy and biological interpretability.
The methodology description needs enhancement in certain areas. While the overall framework is well presented, the justification for choosing the beta-binomial model in the Bayesian framework (lines 356-370) and its hyperparameter determination needs more detailed explanation. The hyperparameter settings (lines 371-374) require clearer justification, especially for the attention heads number in GAT and other optimization parameters.

Validity of the findings

The experimental validation is thorough, with detailed evaluation metrics presented through 5-fold cross-validation. The ablation study results in Table 4 effectively demonstrate the contribution of each component, though statistical significance analysis of these differences would strengthen the findings. The interaction effects between different components (MFSynDCP, GGI, CTF, LSTM) should be explored to provide deeper insights into the model's behavior.
The biological analysis presented in Figure 7(B) shows promise but needs more detailed explanation regarding the criteria for selecting specific drug-target-enzyme interactions and determining edge weights. The biological significance of the identified patterns, particularly in the context of DDI mechanisms, should be more thoroughly discussed.

Additional comments

Several aspects would enhance the manuscript's contribution:
1. The connection between Table 1's drug behavior categories and the final model predictions needs to be more clearly established throughout the results section.
2. Figure 5's ROC curve comparisons would benefit from error bars or confidence intervals to better represent the 5-fold cross-validation results.
3. The data preprocessing section (lines 189-196) should provide more specific details about handling duplicate reports in FAERS and the criteria used for data cleaning.
4. Given the clinical importance of DDI prediction, including detailed case studies of specific drug pairs, particularly those showing high confidence in Figure 6, would demonstrate practical utility.

Reviewer 3 ·

Basic reporting

+ The citation in Lines 98-99 is not formatted correctly: (DEEP LEARNING THEORY AND APPLICATIONS, 2024). Please ensure that all citations follow the required style.
+ Issues in the Introduction Section:
The authors present numerous performance values without providing sufficient context, such as the datasets used, model training procedures, and experimental setups. This lack of detail makes it difficult to compare the performance of different methods, as they were likely measured under varying conditions. To improve clarity and fairness in comparison, the authors should explicitly state the experimental settings for each method mentioned.

Experimental design

1. The Methodology section, particularly the "Data Sources and Preprocessing" subsection, lacks clarity in several areas:
+ FAERS Database:
- What specific information had missing values, and how were these missing values handled?
- The process of "standardizing the formats for drug names and adverse event descriptions" is not explained. Please provide a detailed description of the steps involved.
+ DrugBank Dataset:
- What are the specific chemical properties and interaction data for each drug? Examples should be provided to illustrate these concepts.
- How were these data extracted and processed? A clear explanation of the methodology is needed.
+ STRING and Open Targets Datasets:
- What specific information from these datasets was used, and how was it utilized in the study? Please provide concrete examples to clarify their role in the methodology.
2. Unclear Drug Feature Extraction via MFSynDCP
The description of drug feature extraction using MFSynDCP is insufficient:
- There is no specific function in RDKit that directly converts SMILES strings into molecular graphs. Please clarify the exact process used for this conversion.
- What were the configurations of the Graph Attention Network (GAT) model, and how was it trained?
- What is the format of the features extracted from the GAT model for each drug? A detailed explanation is necessary to understand the feature extraction process.
Similarly, the subsequent sections lack sufficient detail regarding the format of each type of feature. The authors should provide more comprehensive descriptions to ensure reproducibility and clarity.
3. Lack of Clarity in the Curated Dataset
The curated dataset is not described in sufficient detail:
- What specific information does the dataset contain?
- What are the input and output formats for the system?
A clearer description of the dataset is essential for understanding the study's foundation and for enabling reproducibility.
4. Unconvincing Experimental Settings
The experimental settings raise several concerns:
- The dataset was split into a 90:10 ratio, but it is unclear whether any drugs appear in both the training and testing sets. If so, this could lead to data leakage and biased results.
- Was the experiment performed multiple times to ensure robustness?
- The use of cross-validation is not convincing, as a drug may appear in multiple folds, potentially skewing the results.
The authors should address these issues to strengthen the validity of their experimental design.

Validity of the findings

The most significant concern with this paper is the absence of the COMPLETED code and the curated dataset. The authors only provided links to multiple databases, which is insufficient for reproducing the study. Without access to the FULL code and the curated dataset, it is impossible to verify the correctness and reproducibility of the results. This omission severely undermines the credibility of the study. The authors must provide the code and the curated dataset to address these concerns and allow for independent validation of their work.

Additional comments

While the paper addresses an interesting and relevant topic, it suffers from several critical issues, including unclear methodology, insufficient experimental details, and the lack of code and curated datasets. I recommend MAJOR revisions (or REJECT) before considering this paper for further evaluation.

---

## Round 0.2 · Major Revisions

Please take into account the reviewers' feedback and make the necessary revisions to the manuscript, particularly addressing the concerns related to the reproducibility of the results.

Reviewer 1 ·

Basic reporting

The novelty and contribution of the proposed method compared to deep learning-based approaches have become clearer. Citations to prior studies have been added.

Additionally, the differences between the proposed method and existing methods such as MDF-SA-DDI, MATT-DDI, and SubGE-DDI have been added, making the novelty of the proposed method easier to understand.

Experimental design

In the additional experiments, it is shown that the proposed method outperforms the existing methods in each fold of the cross-validation, further strengthening the claims of the paper.

Validity of the findings

no comment

Additional comments

no comment

Reviewer 2 ·

Basic reporting

The authors have addressed my previous concerns effectively. While the paper is largely ready for publication, I recommend the following minor revisions:
1. The model is referred to as both "DABI-DDI" and "DABI-RNN" at different points. Standardize terminology throughout.
2. Several in-text citations have formatting issues (e.g., line 155 "particLin et al. (2023)ularly")
3. The garbled character "¿" appears in some formulas and needs to be checked and restored to the correct mathematical symbol.
4. Line 316: "datseset" should be "dataset". It is recommended to check the full text to avoid similar spelling errors.
5. Multiple instances of "bayesian" not capitalized as "Bayesian".
6. DDI defines acronyms several times in the text. It is recommended that the acronyms be defined throughout the text and the acronyms should be defined at the first occurrence and used elsewhere.
7. Line 508, Biological Attribute Network should be "biological attribute network"

Experimental design

no comment

Validity of the findings

no comment

Additional comments

no comment

Reviewer 3 ·

Basic reporting

There are significant concerns regarding the reproducibility of the results. While the authors have provided a Zenodo link for data and code sharing, the current implementation does not fully meet the requirements for ensuring reproducibility. See below:

Experimental design

1. Incomplete Documentation for Reproducibility
- The Zenodo link should be explicitly mentioned in the manuscript to ensure accessibility for readers.
- The files stored on Zenodo appear disorganized and unclear, making it difficult to reproduce the results.

2. Dataset Preparation and Documentation
- The authors must provide a curated version of the dataset in a structured format.
- A detailed document should be included, outlining the steps to generate the dataset from the original data sources to ensure transparency and reproducibility.

3. Code Execution and Model Training Details
- A step-by-step guide on how to run the code, train the model, and test the model is necessary.
- This documentation should include system requirements, dependencies, hyperparameters, and expected outputs to facilitate smooth replication of the study.

Validity of the findings

There are major concerns as listed above.

Additional comments

While the study contributes to the field, the lack of clear and structured documentation severely limits its reproducibility. The authors are strongly encouraged to curate their dataset, provide step-by-step instructions, and organize their Zenodo repository before resubmission. Addressing these concerns will significantly improve the quality and impact of the manuscript.

---

## Round 0.3 · Minor Revisions

Please address the comments about formatting from Reviewer 2.

Reviewer 2 ·

Basic reporting

The authors have addressed most of my previous concerns, but there are two minor issues that need to be addressed. Perhaps the authors misunderstood my concerns.
1. Regarding previous question 5, I meant to use "Bayesian" uniformly and capitalize the first letter of proper nouns (please check on lines 726, 727, 732, 742, 770, 772).
2. Regarding previous question 6, I meant that abbreviations only need to be defined when they first appear. For example, drug-drug interactions (DDIs) are defined in line 78, but the authors use the full name in lines 104, 216, 263, 300, 302, 304, 399, 429, 775, and 908, and repeat the definition of abbreviations in lines 243, 289, 321, 460, 485, 533, 554, and 889. The same should be true for other abbreviations.
After addressing the above issues, I think the paper can be accepted.

Experimental design

no comment

Validity of the findings

no comment

Reviewer 3 ·

Basic reporting

The authors have addressed my comments in the previous reviewing round.

Experimental design

No comment.

Validity of the findings

No comment.

---

## Round 0.4 · accepted · Accept

Authors have addressed all of the reviewers' comments and the manuscript can be accepted for publication.